# Peer review of "Phylogenetics and Biogeography of Lilium ledebourii from the Hyrcanian Forest"

_diversity, doi:10.3390/d14020137_

Round 1

Reviewer 1 Report

I have carefully read MS which was submitted for consideration in the Diversity MDPI. The genus Lilium L. includes approximately ca. 100 species distributed mainly in temperate regions throughout the northern hemisphere. Infrageneric classification of Lilium is the subject of considerable discussion and the number of sections used differs depending on the author.

The present paper analyses the phylogenetic relationships of Lilium ledeburii based on a phylogenetic analysis of sequence variation of the nuclear ribosomal ITS region and three plastid DNA markers, i.e. matK, trnL-F and trnH-psbA. It aims at a better understanding of the phylogenetic position of the Iranian lily.

The paper is in general well written, logically structured, well-illustrated and easy to understand. It also addresses a subject that is of great interest in the scientific community. The title clearly describes the contents of the paper. The abstract is well written. It encapsulates the entire study (a bit of introduction, aim, result and outcome). The introduction is well written as it gives a good background of the research in question. Also, the aim of the study is evident in the beginning and concluding parts. I believe that the Materials and Methods section is well structured and scientifically sound. The results are well presented, however, some figures need to be improved. Literature reviews in the discussion section of the manuscript are very good.

The current version of the manuscript suffers from several weakness, as detailed below:

Figure 1. This figure is unclear, please correct or prepare a new figure (please consider changing graphic design).

Figure 3. The Latin names of the species on the right side of the figure should be italicized, it looks unprofessional in the current version.

Line 236: Please change "(L. ledeburri)" to (L. ledebourii)

Discussion: What is the opinion of the authors on the classification system of Lilium ledenburii based on the results of morphological and anatomical research? You may want to consider adding a short paragraph at the end of the "Discussion" chapter to address this problem. Of course, if detailed information is available on this.

Author Response

Dear Reviewer

Thank you very much for your helpful suggestion and try to consider all in the text.

Best regards,

Hamed Yousefzafdeh

Reply to reviewer:

Reviewer 1:

C1: Figure 1. This figure is unclear, please correct or prepare a new figure (please consider changing graphic design).

R1: New figure was replaced.

C2: Figure 3. The Latin names of the species on the right side of the figure should be italicized, it looks unprofessional in the current version.

R2: New figure was replaced.

C3: Line 236: Please change "(L. ledeburri)" to (L. ledebourii)

R3: Done.

C4: Discussion: What is the opinion of the authors on the classification system of Lilium ledenburii based on the results of morphological and anatomical research? You may want to consider adding a short paragraph at the end of the "Discussion" chapter to address this problem. Of course, if detailed information is available on this.

R4: A short paragraph as final conclusion was added in the end of discussion.

End.

Reviewer 2 Report

Dear authors, I congratulate you on this very interesting article on a little known species endemic to an area of great interest for its biodiversity as the caspian forests of northern Iran (that I also had the opportunity to study, albeit superficially, in the field).

I consider the work well structured and clear in its presentation. Your phylogenetic analyzes are of great interest and add new data not only for L. ledeburii but for the entire genus Lilium. The only flaw in my opinion is the lack of in-depth study on the taxonomy of the species from a morphological point of view. As I wrote in the comments in the attached text, adding a morphological comparison of L. ledeburii with the species that are closest according to your phylogenetic analyzes would add further value to your article.

Author Response

Dear Reviewer

Thank you very much for your helpful suggestion and try to consider all in the text.

Best regards,

Hamed Yousefzafdeh

Reviewer 2:

C1: The only flaw in my opinion is the lack of in-depth study on the taxonomy of the species from a morphological point of view. As I wrote in the comments in the attached text, adding a morphological comparison of L. ledeburii with the species that are closest according to your phylogenetic analyzes would add further value to your article.

R1:  Yes, this is very good idea, but unfortunately we did not measured morphological characteristics during the study, but the reviewers suggestion will be considered in the future our research on this species.

End.

Reviewer 3 Report

Dear authors,

I have thoroughly read the entire manuscript. Although the idea behind this research is valid, in my opinion, there are many major drawbacks that must be addressed. Throughout the manuscript, you struggle to somehow present L. ledeburii as “a missing link” between Asian and European species. Unfortunately, I do not see any evidence for such a conclusion and I think that you have over-interpreted your results, a lot. First, your analysis is based on only two regions, one nuclear and another plastid. I do not think this is enough, especially if we know that nuclear and plastid genomes have independent evolution. Second, you mentioned there are ca. 100 Lilium species. How many of them have you included in your analysis? And how many of the Middle East species were included? (btw. these information should be clearly stated in the manuscript). And then you conclude that  “our inclusion of Iranian samples here looks critical, as it supports an expansion and secondary diversification in western Asia (i.e. region C) before further range expansion and diversification towards Europe (i.e. region D).” Do you think that if other  Lilium species from the Middle East and the Caucasus would be included in the analysis, the L. ledeburii would still be this “missing link”? Perhaps it would, but on this level, it is not possible to say it is. In essence, you have copied the research of Gao et al. (2013), but you included substantially fewer species and added this narrow endemic L. ledeburii.  Also, the manuscript by Gao et al. was published almost a decade ago, but during the last ten years, the methodology advanced a lot. For instance, look at Kim et al. (Kim, H.T.; Lim, K.-B.; Kim, J.S. New Insights on Lilium Phylogeny Based on a Comparative Phylogenomic Study Using Complete Plastome Sequences. Plants 2019, 8, 547). Also, in the Introduction section, you clearly defined two goals of your research. In the Discussion section, you dealt with your first goal (i.e., phylogenetic positioning of L. ledeburii) in three lines (236-238) and if I am not missing something, you do not address your second goal (i.e., the divergence times of Iranian Lily from other lilies) with a single word. Indeed, you discuss the divergence times of different groups of species, but nothing more.

Among several other very important manuscripts for this topic, you also did not mention this one: Salehi, M., Hatamzadeh, A., Jafarian, V. et al. New molecular record and some biochemical features of the rare plant species of Iranian lily (Lilium ledebourii Boiss.). Hortic. Environ. Biotechnol. 60, 585–593 (2019). In addition, you mentioned Ghanbari et al. (2018) in an irrelevant context, but one of the topics of his paper was also the phylogenetic positioning of L. ledebourii.

After thoroughly reading this paper (some parts for several times), I think that based on the obtained results, it is not ok to discuss anything more than the phylogenetic position of the studied species within the genus (although this was done to some extent in earlier researches). The global biogeography of the lilies, based on the matK and ITS regions, was already done (Gao et al.2013) on a much greater sample set which means that the results there obtained were for sure more reliable than yours. In addition, Salehi et al. (2019) already analyzed the phylogenetic positioning of the species by using the ITS sequence. We all tend to see in our results spectacular things, but usually, this is only an illusion. When (and if) you will try to improve your manuscript, try to be more realistic and not over-enthusiastic regarding the obtained results.

Below you can find my more specific comments.

With best regards.

L2 and throughout the text: is it L. ledeburii or L. ledebourii. Ctrl+F reveals that 13 times it is ledebourii, and eight times it is ledeburii.

L21: remove either “endangered” or “rarest”

L43: it should be “…plant species.”

L44: reference needed. Debated by who?

L48-49: it should be”... only limited information is available regarding its phylogenetic position.”

L54-56: if you say “molecular phylogenetics”, then it is not necessary to add “…DNA barcoding based on nucleotide sequences from variable chloroplastic and nuclear loci…”.

L61: here you refer to a manuscript (Pelkonen and Pirttila) published in the extra low quality, dubious journal which is not being published anymore. But at the same time, some other very important manuscripts in the field are not being mentioned. You should search the available literature regarding Lilium phylogeny more thoroughly.

L65: a narrow endemic species can hardly “presents a critical phylogenetic and biogeographic position among” anything. It is not impossible, but usually, this is not the case. Try to avoid using phrases like “critical…position”.

L68: correct “Liliy”

L73 – M&M section: you wrote that a “total genomic DNA was extracted from at least ten individuals from each population…” (L76-77). That means that you have analyzed at least 40 samples of the studied species. It is very unclear where are these individuals in your results. Then all of the sudden, “samples T-1 – T-8 emerged” as “samples from Iran” (L162-164). But again, “sample T-7 formed a well-supported subclade with L. ledebourii…” (L162-163) which adds additional confusion to already very confusing text. Is not T-7 L. ledebourii itself?

L80, Figure 1: this geographic map was not the best pick. I think there are so many more appropriate/nicer maps that can be used for this purpose. I assume that “Samples” are actually “locations of sampled populations”, right? Why are these photos of plants scattered across the map? Would not it be more appropriate to show them outside the map or as a separate figure? And what exactly are “rare habitats” in this figure? Also, remove “part of Iran”.

subsection 2.2. was the sequencing done from both directions or just in one direction? Because if it was done in one direction, the results might be unreliable. This is a very important issue!

L134-135: when mentioned for the first time, used software or approaches or whatever should be written in long-form, and not acronyms. Bayesian binary MCMC (BBM), Reconstruct Ancestral State in Phylogenies (RASP), etc.

L179-180: if trnH-psbA and trnL-F regions were not informative, then all the results and parts of the manuscript that are somehow linked to these regions should be removed. For instance, it is confusing to see these regions in Tables 1 and 2 if they have nothing to do with any of the obtained results and conclusions.

However, in L56-59 you wrote: “In particular, the nuclear rDNA-internal transcribed spacers (ITS), and chloroplastic rbcL, matK, psbA-trnH, trnL-F and trnH have been widely used to identify and analyze phylogenetic relationships among plant species [14, 15], including in the genus Lilium [16-24].” So how is it possible that they are not informative in your study? I will not go through all these references you mentioned to see which regions were used for the analyses, but it seems that something is wrong here. Also, two years ago a paper was published which offered a solution for this problem (Kim HT, Lim KB, Kim JS. New Insights on Lilium Phylogeny Based on a Comparative Phylogenomic Study Using Complete Plastome Sequences. Plants (Basel). 2019;8(12):547)).

subsection 3.2.: it is not needed to comment on positions of individual differences among samples. The results from statistical analyses are to comment, as all detected differences among samples are parts of these results.

L192, Figure 2. These trees are very blurry and hard to read. Perhaps the review version of the manuscript is of low resolution, if not, please provide figures of higher resolution.

L215. Although it is obvious from the figure, parts (A) and (B) are not marked in the figure. The figure is hard to understand, as comprehensive explanations should be added. Please, explain the legend on the left and the pies on the right. Increase the size of the World map, and perhaps make a separate Figure from it.

L277. Here is for the third time mentioned that the Hyrcanian forest is a hotspot of biodiversity. Ok. And from this “biodiversity hotspot”, you analyzed a single species. I do not see how is it possible to conclude that “this analysis of Iranian samples … illustrates the role the northern forest of Iran (Hyrcanian forest) as a hotspot of biodiversity of international importance”. (L276-278)

Author Response

Dear Reviewer 3

Thank you very much for your helpful suggestion. We have improved the scientific issues, the introduction, the discussion, and reanalysis the data, based on your constructive suggestions. All of these make the manuscript more accurate, more exquisite, and more reasonable. We have replay your questions and suggestions one by one as follow. Please check the whole paper again. Thanks again.

Best wishes,

Corresponding author on behalf of all the authors

Hamed Yousefzafdeh

Reviewer 3:

General comments:

Comment 1: Throughout the manuscript, you struggle to somehow present L. ledebourii as “a missing link” between Asian and European species. Unfortunately, I do not see any evidence for such a conclusion and I think that you have over-interpreted your results, a lot.

Reply 1: Thank you for your suggestions. We used a lot of words to emphasize the importance of L. ledebourii. Now, we also think it is inapposite to excessive description the importance of this species. There are more than 100 lily species distributed in the Northern Hemisphere, we just want to fill the blank of L. ledebourii and rediscuss the biogeography of this genus after added this species from Western Asia. You will see we have improved the whole manuscript based on your this suggestions in the following parts.

 Comment 2: First, your analysis is based on only two regions, one nuclear and another plastid. I do not think this is enough, especially if we know that nuclear and plastid genomes have independent evolution.

Reply 2: Thanks. We understand your concern and confusion. You are right that our analysis is based on only two regions. First, please allow us to state why we chose only two regions. Sequencing is advancing rapidly, while it is too expensive and difficult for us without these technologies in Iran with all kinds of restrictions from outside. Second, we just want to explore the phylogeny and biogeography of L. ledebourii in this paper. Thus, finally, we decided to use the most common markers that there are a lot of public data in the NCBI. Only in this way, we could resolve the phylogeny and biogeography of L. ledebourii more reasonable and correctly. Third,  we know nuclear and plastid genomes have independent evolution, thus we built the phylogeny tree separately based on ITS and matK. There is no problem to do the biogeographic analysis with the combined data with a lot of this kind of study. Finally, I use an example (recent publication on Systematic Biology, IF=15.683) to dispel your doubts for our analysis methods as follow.

Zhang et al. (2022), Fossil-informed models  reveal a Boreotropical origin and divergent evolutionary trajectories in the Walnut family (Juglandaceae). Systematic Biology, 71(1), 242–258.

In this paper, they only used four plastid genes (matK, atpB–rbcL, psbA–trnH, and trnL–trnF) and one nuclear ribosomal DNA (ITS).

Comment 3: Second, you mentioned there are ca. 100 Lilium species. How many of them have you included in your analysis? And how many of the Middle East species were included? (btw. these information should be clearly stated in the manuscript).

Reply 3: Thank you for your suggestions. We have added all the information in the materials and methods part (2.3 phylogenetic analysis and ITS2 secondary structure) as follow:

“There are totally eight individuals of L. ledebourii were used in the phylogenetic analysis in this study. The analysis based on ITS sequences used 35 Lilium species and taken five Tulipa species as outgroups. The analysis based on matK sequences used 21 Lilium species and taken two Tulipa species as outgroups.”

We have added all the information in the materials and methods part (2.4 Divergence time estimate and biogeographic analysis) as follow: “To estimate the divergence time of Lilium and reconstruct the ancestral geographical range of this genus, we used 31 Lilium species covering its entire distribution area.”

Comment 4: And then you conclude that “our inclusion of Iranian samples here looks critical, as it supports an expansion and secondary diversification in western Asia (i.e. region C) before further range expansion and diversification towards Europe (i.e. region D).” Do you think that if other Lilium species from the Middle East and the Caucasus would be included in the analysis, the L. ledeburii would still be this “missing link”?

Reply 4: Thank you. We tried to collect sequences of all species that has recorded their ITS and matK regions in NCBI. Of course, we also regarded to have species from different defined botanical section for the genus Lilium and have samples from the throughout distributed regions of this genus. We used the species that have recorded their both ITS and matK sequencing and due to there not justified for many species, they are excluded from the further investigation. In Caucasus, Lily species have the sequence data of ITS and not recorded data for matK region. Thus, we improved in the discussion as bellow: “This pattern also contrasts with the proposal of Ikinci et al. [49] that European species derive from multiple routes, with L. martagon first colonizing Europe, whereas a second route later gave rise to L. bulbiferum and further diversification of all other European species.”

Comment 5:  In essence, you have copied the research of Gao et al. (2013), but you included substantially fewer species and added this narrow endemic L. ledeburii.  Also, the manuscript by Gao et al. was published almost a decade ago, but during the last ten years, the methodology advanced a lot. For instance, look at Kim et al. (Kim, H.T.; Lim, K.-B.; Kim, J.S. New Insights on Lilium Phylogeny Based on a Comparative Phylogenomic Study Using Complete Plastome Sequences. Plants 2019, 8, 547).

Reply 5: Firstly, we want to declare that we added the species L. ledebourri we focused on haven’t included in Gao et al., 2013. In this study, we added this species and clarified the phylogenetic position of this species.

Secondly, we collected four populations of this species to support our study.  After the phylogeny, we focused on divergence time of L. ledebourri in the Middle East.

As we explained in several parts of the manuscript and the title of manuscript that our research focused on the Lilium ledebourii. We have mentioned that   the distribution of this species is unknown. During this decade we succeed to detect more populations from this species in the Hyrcanian forest, while only one population of this species detected before 2000.

On the other side, we agreed that the methodology advanced a lot followed with the genome time arriving. Thus, we don’t need always cost so much (genomic data) to resolve the questions that we could use DNA Barcoding markers. It means that we don't have to shoot the mosquitoes with cannons.

Also, please see again the biogeography results of Gao et al (2013) paper; They constructed and interpreted biogeography analysis based on ITS and matK regions separately; While we reconstructed and concluded our biogeography analysis by combined data (matK and ITS sequences data) to find our question when Lilium ledebourri diverged from the other Lily and how reach to Hyrcanian forest.

Comment 6: Also, in the Introduction section, you clearly defined two goals of your research. In the Discussion section, you dealt with your first goal (i.e., phylogenetic positioning of L. ledebourii) in three lines (236-238) and if I am not missing something, you do not address your second goal (i.e., the divergence times of Iranian Lily from other lilies) with a single word. Indeed, you discuss the divergence times of different groups of species, but nothing more.

Reply 6: In discussion, we divided this part to two sections and one section dedicated to “Biogeography of Lilies from Iran and the Hyrcanian forest”.  We discussed the biogeography of about Iranian Lily. We then reanalysis the biogeography part and rewrite the discussion part of biogeography.

The new discussion is as follow:

“4.2. Biogeography of Lilium emphasis onWest Asia

Even Lilium is one of the most famous genera all over the world, while the bio-geographic study of this genus is limited [5]. Gao et al. (2013) tried to analysis the evolutionary events in Lilium. While due to the sampling, this study only focuses on the biogeography of Q-T plateau and the Hengduan Mountains. In this study, we try to cover the distribution area of the genus to explore the biogeography of this genus, es-pecially adding some species from West Asia.

Biogeographic reconstructions using S-DIVA indicated an ancestral origin of Lil-ium in Eastern Asia or the extensive area of East-Central-West Asia during the Eocene around 50 MYA (95% HDP: 68.8 – 36.8). This contrasts with prior dating based the plastid dataset that concluded on a last common ancestor of Lilium some 13.19 MYA [5]. Thus, our results suggest that Lilium may be origins early in the Pan-Asian region. The genus began to out of Asia during the Eocene. Two early radiation events from the ancestral area have particularly supported independent colonization toward Western Asia and towards North America. This event may be profit from the warm climatic conditions on this Geologic period. This is coherent with prior hypotheses of migration for diverse plants and animals between the Old and the New Worlds through either the North Atlantic Land Bridge (NALB) or Beringia [47], rather than the suggestion in some literatures that the Himalayas were the center of origin of the genus Lilium [5, 6]. This pattern also contrasts with the proposal of Ikinci et al. [48] that European species derive from multiple routes, with L. martagon first colonizing Europe, whereas a second route later gave rise to L. bulbiferum and further diversification of all other European species. Lilium species of Iran (Western Asia) indeed involved radiation of ancestors from East Asia to Europe, with early colonization of Western Asia from East Asia (50-40 MYA) followed by additional events some 20 MYA and 10 MYA. In particular, three radiations were apparent between Western Asia and Europe in our analysis, supporting dispersal from Western Asia to Europe during the Oligocene (28 MYA) and again during the Miocene (18 MYA).

Due to the limited impact of the ice ages during the Pleistocene, it indeed hosts an impressive number of endemic and relict species [49]. Our analysis showed that the rare Lilium ledebourii is such a relict species that currently scattered among small pop-ulations in steep cliffs above the tree line. These populations seemingly represent sur-vivor of previously large populations having migrated to the area before the ice age. Accordingly, northern forests of Iran likely acted as a refugia for such species. The presence of tree species from typically high latitudes such as Betula pendula and Sorbus aucaparia in the Hyrcanian forest further corroborates the case of L. ledebourii.

A non-mutually exclusive hypothesis is that the north of Iran represent a path for migration of species from East Asia to Europe during periods of drastic environmental changes. Khalilzadeh et al. [50] indeed concluded that the north west of Iran (specifi-cally Southwest Caspian Sea) is a major contact zone between Asian and European clades of Wild Boar and biogeographic inferences on several plant species [51,52] supported the north of Iran (Hyrcanian forest) as a main corridor for plant migration between the East and the West of the Eurasian continent. As expressed by Manafzadeh et al. [53], the Irano-Turanian floristic region can serve as a ‘donor’ of xerophytic taxa to ‘recipient’ neighboring regions, including the Mediterranean floristic region. Addi-tional studies appear necessary to accurately quantify the refugial vs corridor role of the Hyrcanian forest.

In Conclusion, the phylogenetic position and biogeography of endemic lily species in the Hyrcanian forest were studied using barcoding technique. The phylogenetic tree utilizing ITS rather than matK produced results that were more consistent with mor-phological classification and placed L.ledebourii in a phylogenetic group with other Caucasian species and section Liriotypus. Divergence time of L.ledebourii determined that L.ledebourii diverged from radiation of ancestors in East Asia, and then formed an West-Central Asia and Europe lineage during the end of Eocene. Accordingly, the north of Iran appear to act as long-term persistence and migration of Lily species from Asia to Europe.”

Comment 7: Among several other very important manuscripts for this topic, you also did not mention this one: Salehi, M., Hatamzadeh, A., Jafarian, V. et al. New molecular record and some biochemical features of the rare plant species of Iranian lily (Lilium ledebourii Boiss.). Hortic. Environ. Biotechnol. 60, 585–593 (2019). In addition, you mentioned Ghanbari et al. (2018) in an irrelevant context, but one of the topics of his paper was also the phylogenetic positioning of L. ledebourii.

Reply 7: We have added this reference in the introductions and give an explain why we do this again. Please see bellow: “This species is attractive and widely used in ornamental breeding programs, and some biochemical features were also study [18, 19]. Limited information is available regarding its phylogenetic position, one work based on the nuclear rDNA-internal tran-scribed spacers (ITS) and another based on the 5.8S ribosomal DNA sequence recently [19, 20]. However, due to the incomplete sampling on the distribution range of L. ledebourii and single sequence, its phylogenetic position is still unclear.”

Comment 8: After thoroughly reading this paper (some parts for several times), I think that based on the obtained results, it is not ok to discuss anything more than the phylogenetic position of the studied species within the genus (although this was done to some extent in earlier researches). The global biogeography of the lilies, based on the matK and ITS regions, was already done (Gao et al.2013) on a much greater sample set which means that the results there obtained were for sure more reliable than yours. In addition, Salehi et al. (2019) already analyzed the phylogenetic positioning of the species by using the ITS sequence. We all tend to see in our results spectacular things, but usually, this is only an illusion. When (and if) you will try to improve your manuscript, try to be more realistic and not over-enthusiastic regarding the obtained results.

 Reply 7: Thank you for your final suggestions. We have improved the whole paper based on your suggestions.

(1) The global biogeography of the lilies, based on the matK and ITS regions, was already done (Gao et al.2013) on a much greater sample set which means that the results there obtained were for sure more reliable than yours.

Reply (7.1): Yes, Gao et al., 2013 have done the global biogeography of lilies. While, we have make some significant improvement based on their results. First, according to the improved methods, we marge the matK and ITS together to build the phylogenetic tree. Secondly, we add the L. ledebourii the only species from North Iran.

(2) In addition, Salehi et al. (2019) already analyzed the phylogenetic positioning of the species by using the ITS sequence. We all tend to see in our results spectacular things, but usually, this is only an illusion.

Reply (7.2): Yes, there are two papers related the phylogenetic position before, while these studies have some defect as we discussed in the introduction:“This species is attractive and widely used in ornamental breeding programs, and some biochemical features were also study [18, 19]. Limited information is available regarding its phylogenetic position, one work based on the nuclear rDNA-internal tran-scribed spacers (ITS) and another based on the 5.8S ribosomal DNA sequence recently [19, 20]. However, due to the incomplete sampling on the distribution range of L. ledebourii and single sequence, its phylogenetic position is still unclear.”

Special comments

C8: L2 and throughout the text: is it L. ledeburii or L. ledebourii. Ctrl+F reveals that 13 times it is ledebourii, and eight times it is ledeburii.

R8: Corrected.

C9: L21: remove either “endangered” or “rarest”

R9: Done.

C10: L43: it should be “…plant species.”

R10: Done.

C11: L44: reference needed. Debated by who?

R11: “Debated” removed.

C12: L48-49: it should be”... only limited information is available regarding its phylogenetic position.”

R12: Done.

C13: L54-56: if you say “molecular phylogenetics”, then it is not necessary to add “…DNA barcoding based on nucleotide sequences from variable chloroplastic and nuclear loci…”.

R13: removed “molecular phylogenetics”.

C14: L61: here you refer to a manuscript (Pelkonen and Pirttila) published in the extra low quality, dubious journal which is not being published anymore. But at the same time, some other very important manuscripts in the field are not being mentioned. You should search the available literature regarding Lilium phylogeny more thoroughly.

R14: This is not a scientific question. How do you evaluate the published paper? We just referred this paper with some information useful for our study. Secondly, we checked all the publications related to lily again. We think we have referred all the publications important for our study.

C15: L65: a narrow endemic species can hardly “presents a critical phylogenetic and biogeographic position among” anything. It is not impossible, but usually, this is not the case. Try to avoid using phrases like “critical…position”.

R15: Corrected.

C16: L68: correct “Liliy”

R16: Corrected.

C17: L73 – M&M section: you wrote that a “total genomic DNA was extracted from at least ten individuals from each population…” (L76-77). That means that you have analyzed at least 40 samples of the studied species. It is very unclear where are these individuals in your results. Then all of the sudden, “samples T-1 – T-8 emerged” as “samples from Iran” (L162-164). But again, “sample T-7 formed a well-supported subclade with L. ledebourii…” (L162-163) which adds additional confusion to already very confusing text. Is not T-7 L. ledebourii itself?

R17: As you now, we used DNA Barcoding markers for this study.  For phylogenetic work focus on genus level, normally, one or two individuals for each species is enough. To provide more information about L. ledebourii, we used two or three individuals for each population, covered four populations.  We also list some examples as follows to prove our methods.

1- Liu, j., Moller, M., GAO, LM. Zhang, DZ and LI, DZ (2011) DNA barcoding for the discrimination of Eurasian yews (Taxus L., Taxaceae) and the discovery of cryptic species. Molecular Ecology Resources 11: 89–100

2- Liu, J.,Proven , J., Gao, LM and  Li, DZ (2012)Sampling Strategy and Potential Utility of Indels for DNA Barcoding of Closely Related Plant Species: A Case Study in Taxus. International Journal Molecular Science 13:8740-8751

C18: L80, Figure 1: this geographic map was not the best pick. I think there are so many more appropriate/nicer maps that can be used for this purpose. I assume that “Samples” are actually “locations of sampled populations”, right? Why are these photos of plants scattered across the map? Would not it be more appropriate to show them outside the map or as a separate figure? And what exactly are “rare habitats” in this figure? Also, remove “part of Iran”.

R18: This figure was changed. Figure legend also was rewritten.

C19: subsection 2.2. was the sequencing done from both directions or just in one direction? Because if it was done in one direction, the results might be unreliable. This is a very important issue!

R19: We sent the PCR product to the company and ordered to sequence them in two directions.

C20: L134-135: when mentioned for the first time, used software or approaches or whatever should be written in long-form, and not acronyms. Bayesian binary MCMC (BBM), Reconstruct Ancestral State in Phylogenies (RASP), etc.

R20: Done.

C21: L179-180: if trnH-psbA and trnL-F regions were not informative, then all the results and parts of the manuscript that are somehow linked to these regions should be removed. For instance, it is confusing to see these regions in Tables 1 and 2 if they have nothing to do with any of the obtained results and conclusions.

R21: Table 1 moved to Supplementary file. But, in the research that focused on DNA barcoding techniques, it is usual to compare the species based on nucleotide position, please allow us to put this table in the text(table 1).

C22: However, in L56-59 you wrote: “In particular, the nuclear rDNA-internal transcribed spacers (ITS), and chloroplastic rbcL, matK, psbA-trnH, trnL-F and trnH have been widely used to identify and analyze phylogenetic relationships among plant species [14, 15], including in the genus Lilium [16-24].” So how is it possible that they are not informative in your study? I will not go through all these references you mentioned to see which regions were used for the analyses, but it seems that something is wrong here. Also, two years ago a paper was published which offered a solution for this problem (Kim HT, Lim KB, Kim JS. New Insights on Lilium Phylogeny Based on a Comparative Phylogenomic Study Using Complete Plastome Sequences. Plants (Basel). 2019;8(12):547).

R22: The sentences were modified. Based on literature review two plastid regions (rbcl and matK) are informative and the other mentioned regions (psbA-trnH, trnL-F and trnH) were not informative.

C23: subsection 3.2.: it is not needed to comment on positions of individual differences among samples. The results from statistical analyses are to comment, as all detected differences among samples are parts of these results.

R23: As you have seen, we used four populations of L. ledebourii. It is informative to comment on the phylogenetic relationship among these populations for this species.

C24: L192, Figure 2. These trees are very blurry and hard to read. Perhaps the review version of the manuscript is of low resolution, if not, please provide figures of higher resolution. L215. Although it is obvious from the figure, parts (A) and (B) are not marked in the figure. The figure is hard to understand, as comprehensive explanations should be added. Please, explain the legend on the left and the pies on the right. Increase the size of the World map, and perhaps make a separate Figure from it

R24: Done.

C25: L277. Here is for the third time mentioned that the Hyrcanian forest is a hotspot of biodiversity. Ok. And from this “biodiversity hotspot”, you analyzed a single species. I do not see how is it possible to conclude that “this analysis of Iranian samples … illustrates the role the northern forest of Iran (Hyrcanian forest) as a hotspot of biodiversity of international importance”. (L276-278)

R25: The mentioned sentence was removed.

End.
